# Field Evaluation of a Hemozoin-Based Malaria Diagnostic Device in Puerto Lempira, Honduras

**DOI:** 10.3390/diagnostics12051206

**Published:** 2022-05-11

**Authors:** Gustavo Fontecha, Denis Escobar, Bryan Ortiz, Alejandra Pinto, Delmy Serrano, Hugo O. Valdivia

**Affiliations:** 1Microbiology Research Institute, Universidad Nacional Autónoma de Honduras, Tegucigalpa 11101, Honduras; denis.escobar@unah.edu.hn (D.E.); bryan.ortiz@unah.edu.hn (B.O.); mpinto@unah.edu.hn (A.P.); 2Hospital de Puerto Lempira, Secretaría de Salud de Honduras, Gracias a Dios 33101, Honduras; delmyverosl@yahoo.com; 3Department of Parasitology, U.S. Naval Medical Research Unit 6 (NAMRU-6), Lima 07006, Peru; hugo.o.valdivia.ln@mail.mil

**Keywords:** malaria, Gazelle™, Honduras, diagnostic, nested PCR

## Abstract

The diagnosis of malaria in Honduras is based mainly on microscopic observation of the parasite in thick smears or the detection of parasite antigens through rapid diagnostic tests when microscopy is not available. The specific treatment of the disease depends exclusively on the positive result of one of these tests. Given the low sensitivity of conventional methods, new diagnostic approaches are needed. This study evaluates the in-field performance of a device (Gazelle™) based on the detection of hemozoin. This was a double-blind study evaluating symptomatic individuals with suspected malaria in the department of Gracias a Dios, Honduras, using blood samples collected from 2021 to 2022. The diagnostic performance of Gazelle™ was compared with microscopy and nested 18ssr PCR as references. The sensitivity and specificity of Gazelle™ were 59.7% and 98.6%, respectively, while microscopy had a sensitivity of 64.9% and a specificity of 100%. The kappa index between microscopy and Gazelle™ was 0.9216 using microscopy as a reference. Both methods show similar effectiveness and predictive values. No statistical differences were observed between the results of the Gazelle™ compared to light microscopy (*p* = 0.6831). The turnaround time was shorter for Gazelle™ than for microscopy, but the cost per sample was slightly higher for Gazelle™. Gazelle™ showed more false-negative cases when infections were caused by *Plasmodium falciparum* compared to *P. vivax*. Conclusions: The sensitivity and specificity of Gazelle™ are comparable to microscopy. The simplicity and ease of use of the Gazelle™, the ability to run on batteries, and the immediacy of its results make it a valuable tool for malaria detection in the field. However, further development is required to differentiate *Plasmodium* species, especially in those regions requiring differentiated treatment.

## 1. Introduction

Malaria in Honduras increased by more than 46% between 2020 and 2021 when 1671 cases were reported in the country, with 98% of them concentrated in the department of Gracias a Dios (Personal communication, National Office of the Pan American Health Organization). This increase in malaria cases represents a setback in efforts to achieve elimination [1], which had been commendable before the COVID-19 pandemic [2]. Another recent change in the epidemiology of malaria in Honduras is the increase in the proportion of cases caused by *P. falciparum* versus *P. vivax*, from 29.3% in 2020 to 43.5% in 2021 (Personal communication by the National Malaria Surveillance Laboratory, Health Ministry, Honduras).

Diagnosis and timely treatment are mainstays in malaria control and elimination, preventing local transmission. The diagnosis of malaria in Honduras is mainly based on the detection of the parasite by light microscopy (LM) carried out by expert technicians or by rapid diagnostic tests (RDTs). LM remains the gold standard in many endemic countries worldwide [3] due to its low cost and because it allows the identification of the species and stages of the parasite, in addition to being useful for the assessment of the severity of malaria through the quantification of parasitaemia. However, even under optimal conditions, the detection limit of LM is low (0.001% parasitemia) [4,5]. Furthermore, microscopy is a time-consuming method and relies heavily on skilled technicians. The loss of diagnostic capacity based on microscopy is a common problem in countries where the number of malaria cases has decreased [6,7]. The declining skills of microscopists have negative consequences for diagnosis, such as an underestimation of cases due to low parasite burden among symptomatic and asymptomatic persons [8], misidentification of parasite species [9], or inability to detect mixed infections [10].

On the other hand, RDTs are valuable alternatives to microscopy, especially in remote areas lacking qualified personnel or continuous electrical flow. RDTs detect parasite-specific antigens, are inexpensive and easy to use in the field, and allow immediate treatment after diagnosis. However, the sensitivity of common RDTs is usually lower than that of microscopy, with a threshold of 100–200 parasites per μL of blood [11,12]. Furthermore, RDTs do not indicate parasite burden and may face detection problems when the parasite does not express the target antigen [13,14]. Due to the intrinsic limitations of conventional malaria diagnostic methods and the challenges faced by countries approaching elimination, a large number of new diagnostic methods have been proposed [15,16,17,18,19,20,21,22]. The most sensitive methods (around five parasites/μL of blood) are those based on DNA amplification. However, they require specialized infrastructure and highly qualified technicians, are expensive, labor-intensive, and cannot be implemented at the point of care [23]. Therefore, molecular methods are restricted to research or reference laboratories.

Thus, there is a real need to develop new diagnostic approaches that are easy to implement at the local level, inexpensive, and with a sensitivity at least comparable to the gold standard, LM (below 50–100 parasites/μL of blood), if we want to achieve the United Nations Sustainable Development Goal number 3 for the elimination of malaria [1]. One of the most studied biomarkers at present is hemozoin or malarial pigment. Hemozoin is an inert crystal by-product of hemoglobin degradation, produced by all species of *Plasmodium* spp. within the digestive vacuole. The purpose of hemozoin production is to avoid the toxicity that heme groups would have on the parasite through oxidative stress [24,25]. Hemozoin is a good biomarker because it is generally absent in uninfected individuals, and its blood concentration increases with the progression of malaria [26]. Due to its chemical structure, hemozoin is paramagnetic and optically birefringent [27,28]. These properties have allowed the development of micro and miniaturized devices, extensively reviewed by Baptista et al. [29], to detect malaria using simple and portable instruments.

This study evaluates the performance of Gazelle™ (Hemex Health, Portland, OR, USA), a late-stage device to diagnose malaria by detecting hemozoin. The study was conducted in a setting of moderate endemicity for malaria in Central America, where two species of *Plasmodium* circulate. Gazelle™ was compared with LM and nested PCR.

## 2. Materials and Methods

### 2.1. Study Population

This was a double-blind study evaluating symptomatic individuals suspected of malaria comparing three diagnostic assays: thick blood smear, Gazelle™, and nested PCR. The study was carried out in the department of Gracias a Dios in the Moskitia region between 2021 and the first trimester of 2022. La Moskitia is an isolated territory located in eastern Honduras (Figure 1) that contributed to 98% of national malaria cases last year. Two species of human *Plasmodia* coexist in the region, *P. falciparum* and *P. vivax* [30]. The most common vector species in the country and in La Moskitia is *Anopheles albimanus* [31]. The geographic isolation of the Honduran Moskitia is caused by multiple lagoons and a thick strip of forest that limits land access. Around this system of lagoons of 3700 square kilometers live more than 24,000 people far from each other. The remoteness and the difficulties of access limit the development of the communities with respect to the rest of the country, which is reflected in the low health and education indicators, with a human poverty rate of 53%. The mangrove forest is the dominant coastal ecosystem with more than 23,000 hectares, which privileges fishing as the main means of subsistence and favors the development of malaria vector mosquitoes [32].

### 2.2. Study Population and Ethics

Blood samples were collected from febrile patients who attended routine medical care at the public hospital in the municipality of Puerto Lempira, in the department of Gracias a Dios, located within the region of La Moskitia. Participants resided in one of the following municipalities: Puerto Lempira (91.8%), Ramón Villeda Morales (4%), Ahuas (2.7%), Brus Laguna (0.9%), and Wampusirpi (0.5%) (Figure 1). Patients of both sexes and of all ages were recruited. The patients or their legal guardians were informed of the objectives of the study and signed an informed consent form before collecting the blood samples. There were no exclusion criteria for this study. The study was conducted in accordance with the guiding principles of the Declaration of Helsinki and was approved by the Institutional Ethics Committee of the National Autonomous University of Honduras (CEI-MEIZ/UNAH). The protocol for this study (03-2020/NAMRU6.2018.0002) was reviewed and approved by the Research Administration Program of the Naval Medical Research Unit-6 (NAMRU-6).

### 2.3. Sample Size

No previous studies have tested the sensitivity of Gazelle™ in a low transmission setting similar to this study. Therefore, the sample size was calculated as described by Buderer [33], assuming a sensitivity of at least 55%, a specificity of at least 85%, and a disease prevalence of 1/3 (33.3%) on collected samples, 12% relative precision, and 80% power. This yielded a minimum required sample size of 199 subjects. Further, 10% of the samples were added to cover possible sample losses for any unavoidable reasons leading to a final sample size of 219 participants.

### 2.4. Microscopy

After the physician evaluated the feverish patients suspected of malaria, trained laboratory technicians collected the blood samples in EDTA tubes. According to the Honduran national malaria guidelines, thin and thick blood smears were prepared for microscopic examination [34]. The slides were examined within the first 24 h after the sample was collected. A total of 500 microscopic fields were observed with a 100X objective before reporting negative samples. Positive samples were reported estimating the parasite density through a quantitative method, reporting both asexual and sexual stages of the parasite per 200 leukocytes. The density of parasitemia was classified as described by Alger et al. [35] as high, moderate, or low, depending on the parasite species.

### 2.5. Gazelle™ Device

The Gazelle™ device was handled by a technician in the clinical laboratory of Puerto Lempira hospital. Blood samples were processed at the end of each working day by blinded study microscopists to avoid bias. The Gazelle™ device uses magneto-optical technology to detect hemozoin produced by *Plasmodium* as a by-product of hemoglobin digestion. By exposing the iron contained in the hemozoin to an intense magnetic field, the opacity of the sample changes, and the increase in opacity is indicative of malaria. The Gazelle™ system consists of a portable reading device and single-use disposable cartridges that operate on either battery power or electricity. The cartridges consist of a chamber, where 65 microliters of the diluent provided by the manufacturer were deposited, followed by 30 microliters of patient blood into the diluent. The upper part of the cartridge was assembled in the lower chamber, and once assembled, it was inserted into the reader, and the analysis was started. The results were recorded after at least one minute to determine the presence or absence of malaria. The Gazelle™ system is unable to differentiate between *Plasmodium* species responsible for the infection. The Gazelle™ device can be seen in Figure 2.

### 2.6. DNA Extraction and Nested PCR

A volume of 50 µL of blood was deposited on Whatman filter paper Nº 3 for further DNA extraction. The samples were stored at room temperature in individual sealed plastic bags with desiccant for up to five months before being transported to Tegucigalpa. The DNA was extracted using the AutoMate™ system and the PrepFiler Express Forensic DNA Extraction Kit™ (Applied Biosystems, Waltham, MA, USA) according to the manufacturer’s instructions.

A nested PCR (nPCR) was performed to amplify a segment of the ribosomal 18S gene using the primers and conditions described by Singh et al. [36,37] with modifications. Briefly, both reactions were performed in 50 µL total volume containing 2X master mix Taq polymerase (Promega Corp. Madison, WI, USA) and 2 µL of each primer 10 µM (Table 1). The first reaction included 11 µL of nuclease-free water and 10 µL of DNA. The second reaction included 20 µL of nuclease-free water and 1 µL of the product of the first reaction.

Negative samples were considered negative after only one result, while positive samples were amplified a second time to confirm the result. If a discordant result was obtained between two PCR reactions or between a PCR reaction and microscopy/Gazelle™, the PCR was repeated a third time, and the result was determined by two concordant tests. The third PCR reaction was performed with newly extracted DNA from the filter paper to exclude contamination. Samples with two positive results were analyzed to determine the parasite species through two independent reactions that were carried out in a final volume of 25 µL containing 12.5 µL of 2X master mix Taq polymerase, 1 µL of each primer 10 µM, 9.5 µL of nuclease-free water and 1 µL of the product of the first PCR (Table 1). The analysts of the nPCR were blinded to the malaria status of the samples before performing the technique. Once the result of the nPCR was obtained, the result of the microscopy and Gazelle™ were revealed to decide if it was necessary to repeat the PCR.

The reactions were carried out by an initial denaturation at 94 °C for 4 min, 35 cycles of 94 °C for 30 sec, annealing temperature for 60 sec (Table 1), and 72 °C for 60 s, with a final extension at 72 °C for 4 min. Amplicons were visualized by 2% agarose gel electrophoresis with ethidium bromide. The positive and negative controls were included in each set of reactions.

### 2.7. Turn-Around Time and Cost Analysis

The turnaround time and cost per test were calculated for the thick blood smear, Gazelle™, and nPCR. The cost calculation included consumables and reagents. The cost of collecting the blood sample and preparing the thick smear for microscopy was included, as well as the cost of sampling, cartridges, and buffers for Gazelle™. Filter paper, semi-automatic DNA extraction, amplification reactions, and electrophoresis for PCR were considered. The fixed costs for labor, the laboratory, and the equipment were not included.

### 2.8. Statistical Analysis

The sensitivity (Sen), specificity (Spe), positive and negative predictive values (PPV, NPV) were calculated for microscopy and Gazelle™ using the PCR as the reference assay, where Sen = true positives/total positives ∗ 100; Spe = true negatives/total negatives ∗ 100; PPV = true positives/(true positives + false positives) ∗ 100; NPV = true negatives/(true negatives + false negatives) ∗ 100. The diagnostic accuracy (or effectiveness) was calculated as follows: (true positives + true negatives)/(true positives + true negatives + false positives + false negatives) *100 [38]. The total number of samples were those on which all three assays were successfully performed. Then, 95% confidence intervals were calculated for the sensitivity and specificity of microscopy and Gazelle™.

Receiver operating characteristic (ROC) analysis and areas under the curve (AUC) were carried out to assess diagnostic accuracy and to compare the diagnostic performance of microscopy, Gazelle™, and nested PCR. AUC was interpreted as follows: 0.9–1.0, excellent; 0.8–0.9, very good; 0.7–0.8, good; 0.6–0.7, sufficient; 0.5–0.6, bad; <0.5, test not useful [39].

The Cohen´s kappa coefficient of agreement of microscopy and Gazelle™ using the nPCR as reference was also calculated as follows: k = p_o_ − p_e_/1 − pe, where p_o_ is the relative observed agreement among assays, and p_e_ is the hypothetical probability of chance agreement. In addition, the kappa index between microscopy and Gazelle™ was calculated using microscopy as a reference. The kappa index result was interpreted as follows: values ≤ 0 indicate no agreement and 0.01–0.20 as none to slight, 0.21–0.40 as fair, 0.41–0.60 as moderate, 0.61–0.80 as substantial, and 0.81–1.00 as almost perfect agreement [40]. The McNemar chi-square was calculated to assess statistical differences between the results of Gazelle™ compared to LM.

## 3. Results

We enrolled 220 febrile patients who requested medical care at the Puerto Lempira Hospital. Out of those, 145 (66%) were female, and 73 (33.2%) were minors under the age of 18 years. The average age of the participants was 24.6 years, with an age range of one month to 74 years.

Microscopy diagnosed 50 (22.73%) samples with malaria, whereas Gazelle™ detected 48 (21.82%) positives, and nPCR detected 77 (35%) positives (Table 2). Microscopy detected 30 samples infected by *P. vivax*, 19 by *P. falciparum*, and one sample with mixed infection. Nested PCR revealed 50 samples infected with *P. vivax*, 19 with *P. falciparum*, and eight with mixed infections (Table 2). When comparing the concordance in the identification of *Plasmodium* species between microscopy and nPCR, microscopy misidentified three *P. vivax* infections as *P. falciparum* and missed seven cases with mixed infection (Table 3).

According to microscopy, the median parasite density (parasites/100 leukocytes) was 57, with a range of 0.4 to 223. Parasitaemia was considered high, moderate, or low in 28 (56%), 14 (28%), and eight (16%) patients, respectively. Only one patient showed a parasitemia under 200 parasites/μL and was shown to be negative by Gazelle™. The WHO uses this value in the RDT quality assurance program as a cut-off point to assess the sensitivity of rapid diagnostic tests [41].

In four positive cases of *P. falciparum* detected by microscopy, Gazelle™ did not detect hemozoin (one case with high parasitemia, two moderates, and one low). All four cases were confirmed by nPCR, and accordingly, they were considered false negatives by Gazelle™. In contrast, two positive cases by Gazelle™ were not detected by microscopy. In none of them were the infections were confirmed by nPCR, and therefore they were classified as false positives by Gazelle™. The kappa index between microscopy and Gazelle™ was 0.9216 using microscopy as reference.

### Performance of Microscopy and Gazelle

Using nPCR as the reference test, microscopy had a sensitivity of 64.9% (95%CI: 62.8–67.0%) and a specificity of 100% (95%CI: 98.6–101.4%), while Gazelle™ had a sensitivity of 59.7% (95%CI: 57.9–61.6%) and a specificity of 98.6% (95%CI: 98.2–99.0%) (Table 4). The ROC analysis showed that the area under the curve (AUC) was 0.79 for Gazelle™ and 0.82 for microscopy (Figure 3). The McNemar chi-square test did not show statistical differences between the results of Gazelle compared to LM (*p* = 0.6831).

The predictive values, effectiveness, and kappa index for microscopy and Gazelle™ are also shown in Table 4. According to the McHugh table, the agreement between microscopy and Gazelle™ with respect to nPCR was moderate, and the agreement between microscopy and Gazelle™ was almost perfect.

When comparing the percentage of false negatives for Gazelle™ vs. nPCR by parasite species, Gazelle™ missed 20 of 50 (40%) *P. vivax* infections and 10 of 19 (52.6%) *P. falciparum* infections. If mixed infections are included in the analysis, Gazelle™ missed 21 of 58 *P. vivax*/mixed infections (36.2%) and 11 of 27 *P. falciparum*/mixed infections (40.7%). Due to inconsistencies between two assays, 56 of 220 (25.5%) samples were analyzed more than once by nPCR. Out of those, 43 of 56 (76.8%) samples were positive in two or three consecutive nPCR assays, while 10 of 56 (17.9%) had two positives and one negative result, and 3 of 56 (5.4%) had two negatives and one positive result.

As shown in Table 5, the turnaround time was shorter for Gazelle™ than for microscopy, but the cost per sample was slightly higher for Gazelle™, not considering fixed costs, human resources, and equipment. The turnaround time and costs per sample are considerably higher for nPCR.

## 4. Discussion

In this study, we evaluated the in-field performance of a novel diagnostic device (Gazelle™) based on the detection of hemozoin, comparing it with microscopy and nested PCR. The sensitivity of Gazelle™ was lower but comparable to microscopy (60% versus 65%, respectively), with similar effectiveness and predictive values and a kappa index showing almost perfect agreement between both methods. Other field studies have evaluated the performance of Gazelle™. A study conducted in the Peruvian Amazon that included samples infected by *P. vivax* reported a sensitivity of 88.2% for Gazelle™, slightly lower than the sensitivity of LM, although statistically similar, and using a nested PCR as the gold standard [42]. In the Brazilian Amazon, Gazelle™ and LM were compared using TaqMan qPCR as a reference. Only malaria vivax samples were evaluated, with a sensitivity of 72.1% for Gazelle™ and 75% for microscopy. Gazelle™ sensitivity was 96.2% compared to LM as a reference [43]. In India, Gazelle™ was evaluated in a region where four *Plasmodium* species are endemic. The sensitivity of Gazelle™ was 82.1% compared to nPCR, while the sensitivity of LM was 75%. The sensitivity of Gazelle™ was 97.6% using LM as the reference test [44]. In Papua New Guinea, the sensitivities of Gazelle™ and LM were 78% and 82%, respectively, compared to nested PCR, in a region where three parasite species coexist [45]. Regarding the specificity, our results are similar to those of other studies, approaching 100% in all cases, which shows that Gazelle™ is a reliable tool to differentiate between positive and negative cases.

In general, the results obtained here are consistent with the literature, and the relatively low sensitivity for both Gazelle™ and LM could be attributed to unknown biological characteristics of the parasites circulating in the region or low parasitic burden [43]. Low parasitaemia and submicroscopic infections have been described as the main causes of the low sensitivity of common malaria diagnostic tools [46,47], particularly among asymptomatic individuals and in low malaria-endemic settings [7,48]. In this study, parasitaemia was only calculated for positive samples by microscopy; therefore, it was not possible to determine the parasite burden in samples positive by PCR but negative by microscopy or Gazelle™. However, in 13 of 56 cases (23.2%) in which PCR was performed in triplicate, there were discordant results between the PCRs. This could suggest that the parasitemia of these individuals was around the detection limit of the most sensitive technique (nPCR) and could be responsible for the low sensitivity of Gazelle™. On the other hand, and at the direct suggestion of the manufacturer, in this study, the method of preparing the blood sample deposited in the Gazelle™ cartridge was modified. Previous studies have used 15 µL of blood in 80 µL of the diluent [42,43,44]; instead, in this study, we used 30 µL of blood in 65 µL of diluent. Although doubling the sample volume should have increased sensitivity, this variable could be responsible for an inverse effect. However, further studies will be necessary to evaluate this possibility.

In this study, as well as in previous reports, Gazelle™ detected a few false-positive cases. Arndt et al. showed that past malaria infections are associated with residual levels of hemozoin, especially in individuals infected within the last two weeks [45]. However, this hypothesis cannot be verified here due to the lack of access to the clinical history of the participants. On the other hand, Gazelle™ missed 40% of *P. vivax* infections but 52.6% of *P. falciparum* infections, which could be due to a difference in sensitivity for both species. Some authors have shown that *P. falciparum* infections tend to go unnoticed by magneto-optical detection of hemozoin [45]. Since the concentration of hemozoin in the peripheral bloodstream depends in part on the circulating stages of the parasite, and because ring immature trophozoites predominate in *P. falciparum* infections in the absence of other asexual stages, the amount of hemozoin is lower compared to *P. vivax* infection [49,50].

About the implementation of the technique, Gazelle™ has the advantage of a very fast response time (less than 5 min), which would positively impact timely diagnosis and treatment. In addition, the cost per test is similar to that of microscopy and RDTs, and although an initial investment of USD 700 is required to acquire the device, the amount is justified by the high average life, with a capacity of up to 20,000 tests [42]. Another advantage of Gazelle™ is its portability and the ability to run on battery power and in high temperatures, making it suitable for remote locations in the tropics with limited infrastructure. One drawback is that Gazelle™ requires phlebotomy to collect the venous blood sample, as well as the correct use of a micropipette to dispense the proper volume of blood and diluent into the cartridge. These skills are greater than those required by an RDT but much less than those of an expert microscopist. The subjectivity of microscopy is another limitation overcome with Gazelle™. The relatively little training required to operate Gazelle™ would allow its use to be expanded in remote communities with a trained technician available. This makes Gazelle™ a possible solution to the lack of skilled microscopists in countries approaching elimination. One of the most attractive features of the device is the ability to store data and transmit it electronically over a wireless connection, enabling real-time responses from decision-makers.

The most notable disadvantage of Gazelle™ is its inability to distinguish between the species of the parasite. *Plasmodium vivax* and *P. falciparum* circulate in Honduras, and the national guidelines establish a differentiated treatment with primaquine 0.25 mg/kg for 14 days against *P. vivax* to ensure the elimination of liver stages and 0.75 mg/kg in single-dose against *P. falciparum* on the first day of treatment. In case of mixed infection, the scheme against *P. vivax* is followed [34]. Consequently, the diagnosis of malaria requires mandatory identification of the species of the parasite and the determination of the parasitemia. Including a method such as Gazelle™ in the national malaria program would require that all positive samples be subsequently analyzed by microscopy before the administration of treatment, which would be impractical and increase costs. A new version of Gazelle should detect all parasite species with sensitivity at least equal to that of LM or RDTs.

## 5. Conclusions

This study evaluated the in-field performance of a device based on the magneto-optical detection of hemozoin (Gazelle™), demonstrating sensitivity and specificity comparable to LM. Its short turnaround time, low cost, and ease of use make it a valuable tool for point-of-care malaria detection and an alternative to the lack of trained microscopists. However, further development is required to differentiate the species of *Plasmodium* if the device is to be introduced in places whose regulations establish a differentiated treatment.

## Figures and Tables

**Figure 1 diagnostics-12-01206-f001:**
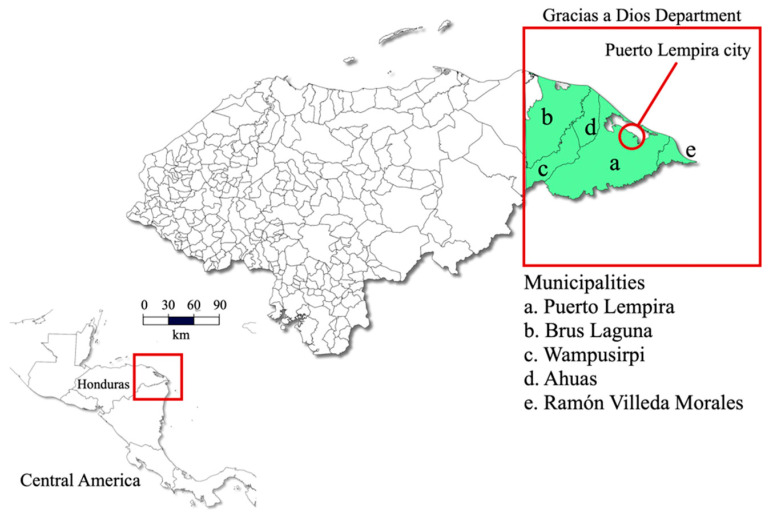
Map of Honduras showing the Department of Gracias a Dios, the city of Puerto Lempira, and the five municipalities of origin of the participants.

**Figure 2 diagnostics-12-01206-f002:**
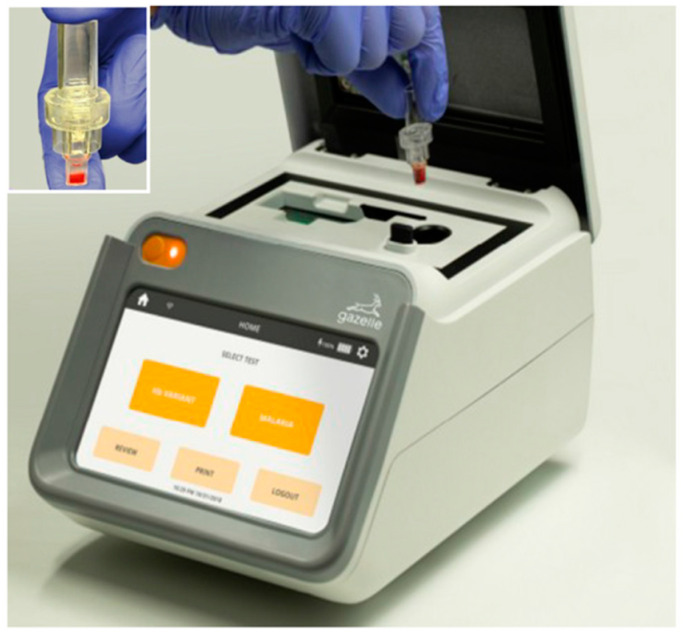
Cartridge loaded with blood sample at the time of insertion into the Gazelle™ device prior to hemozoin detection analysis. Image courtesy of Hemex Health.

**Figure 3 diagnostics-12-01206-f003:**
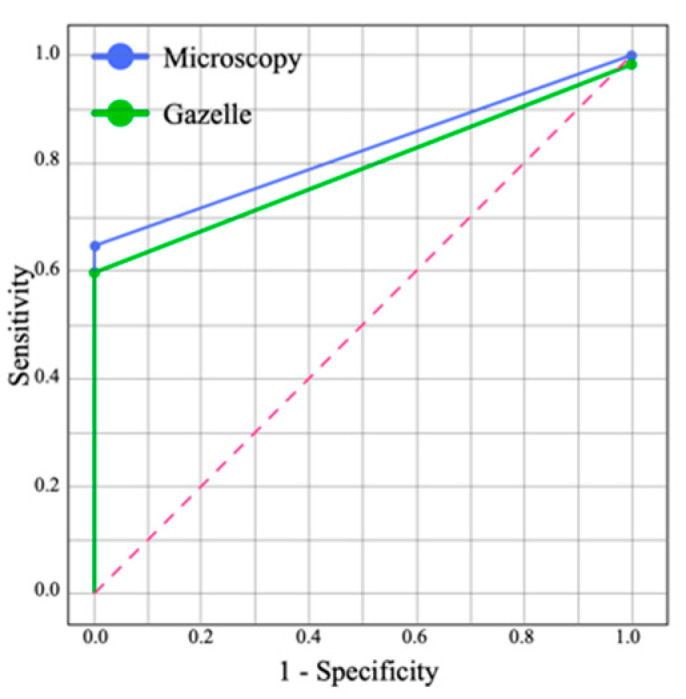
ROC curves for microscopy and Gazelle™ compared with nested PCR as reference test. Gazelle™ presented an AUC of 0.79 (good) and LM an AUC of 0.82 (very good).

**Table 1 diagnostics-12-01206-t001:** Primers and PCR conditions used for *Plasmodium* detection.

PCR Reaction	Primer Name	Primer Sequence (5′-3′)	Annealing Temperature	Amplicon Size (bp)
Genus *Plasmodium* first PCR	rPLU1	TCA AAG ATT AAG CCA TGC AAG TGA	55 °C	
	rPLU5	CCT GTT GTT GCC TTA AAC TYC		
Genus *Plasmodium* nested PCR	rPLU3	TTT YTA TAA GGA TAA CTA CGG AAA AGC TGT	62 °C	240
	rPLU4	TAC CCG TCA TAG CCA TGT TAG GCC AAT ACC		107
*Plasmodium vivax*	rVIV1	CGC TTC TAG CTT AAT CCA CAT AAC TGA TAC	58 °C	
	rVIV2	ACT TCC AAG CCG AAG CAA AGA AAG TCC TTA		205
*Plasmodium falciparum*	rFAL1	TTA AAC TGG TTT GGG AAA ACC AAA TAT ATT	58 °C	
	rFAL2	ACA CAA TGA ACT CAA TCA TGA CTA CCC GTC		

**Table 2 diagnostics-12-01206-t002:** Number of positive and negative samples for malaria according to three diagnostic tests and parasite species identification.

Assay	Positive Samples (%)	Negative Samples (%)	*P. vivax* (%)	*P. falciparum* (%)	Mixed Infections (%)
Microscopy	50 (22.7%)	170 (77.3%)	30 (60%)	19 (38%)	1 (2%)
Gazelle	48 (21.8%)	172 (78.2%)	N.A.	N.A.	N.A.
Nested PCR	77 (35%)	143 (65%)	50 (64.9%)	19 (24.7%)	8 (10.4%)

**Table 3 diagnostics-12-01206-t003:** Concordance in the diagnosis of *Plasmodium* species between microscopy and nested PCR.

	Microscopy			
nPCR	*P. vivax*	*P. falciparum*	Mixed Infections	Total
*P. vivax*	27 (54%)	3 (6%)	0	30
*P. falciparum*	0	12 (24%)	0	12
Mixed infections	3 (6%)	4 (8%)	1 (2%)	8
Total	30	19	1	50

**Table 4 diagnostics-12-01206-t004:** Microscopy and Gazelle™ performance values in relation to the reference method (nested PCR).

	Microscopy	Gazelle
Sensitivity	64.9%	59.7%
Specificity	100%	98.6%
PPV	100%	95.8%
NPV	84.1%	82.0%
Accuracy (effectiveness)	87.73%	85%
Kappa index	0.7065	0.6389

**Table 5 diagnostics-12-01206-t005:** Turnaround time and cost analysis considering reagents and supplies. No equipment no labor costs are included. The turnaround time calculation included the sampling time.

Assay	Cost (USD) per Sample	Turnaround Time (Minutes)
Microscopy	<$1	15–30
Gazelle	$1.25	<5
Nested PCR	$15–20	1020

## Data Availability

Data supporting the conclusions of this article are included within the article and its additional files. The raw data used and/or analyzed during the present study are available from the corresponding author on reasonable request.

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
