# Peer review of "Field Evaluation of a Hemozoin-Based Malaria Diagnostic Device in Puerto Lempira, Honduras"

_diagnostics, 2022, doi:10.3390/diagnostics12051206_

Round 1

Reviewer 1 Report

Author has presented the field evaluation of Gazalle devise for malaria diagnosis.

There is no novelty except the field site.

the sample size calculation is not appropriate, if we assume the sensitivity of any device is 55% then there is no question to evaluate such device as it has no diagnostic value 

the performance of gazalle was compared with microscopy however the main tool for malaria diagnosis in Honduras is RDT. So it is better to compare with RDT too.

PCR detected more cases which were missed by Gazalle or microscopy, what is the treatment given in those cases as all are fever cases.

How author can conclude that Gazelle is more sensitive in P. vivax infection as compared to the P. falciparum as it is not the study design. If author want to compare the species then sample size may calculated accordingly the prevalence of those species in the study area.

Overall manuscript need extensive rewrite in every section.

Author Response

  • There is no novelty except the field site.

Answer: The novelty of this study consists in the evaluation of the Gazelle device in a scenario in which two species of Plasmodia (P. falciparum and P. vivax) cohabit. The only study that evaluated this same variable was carried out in India (Kumar et al 2020), with strains of the parasite with different characteristics from those found in Central America. Other studies have evaluated the device in scenarios where only P. vivax circulates in Brazil (de Melo et al 2021), Peru (Valdivia et al 2021), and Thailand (Orban et al 2021). In such a way we consider that our results provide valuable information if this new technology can be applied in all malaria-endemic regions. In addition, no previous studies were made on the device in such a low transmission setting like Honduras.

  • the sample size calculation is not appropriate, if we assume the sensitivity of any device is 55% then there is no question to evaluate such device as it has no diagnostic value 

Answer. The reviewer needs to take into consideration that no previous studies were made on the device in such a low transmission setting like Honduras. Therefore, establishing a low sensitivity of at least 55% results in the highest sample size for such a validation. 

This approach has been previously performed in the other two Gazelle publications from India and Peru which are published at the Lancet Eclinical Medicine and PLoS One. 

Also, the intend of this study is to evaluate its diagnostic value. It is not up to the author nor the reviewer to affirm its diagnostic value beforehand.

Anyhow, the paragraph has been modified as follows: “No previous studies have tested the sensitivity of GazelleTM on a low transmission setting similar to this study. Therefore, the sample size was calculated as described by Buderer assuming a sensitivity of at least 55%, specificity of at least 85%, a disease prevalence of 1/3 (33.3%) on collected samples, 12% relative precision and 80% power. This yielded a minimum required sample size of 199 subjects. Further 10% samples were added to cover possible sample losses for any unavoidable reasons leading to a final sample size of 219 participants”.

  • the performance of gazalle was compared with microscopy however the main tool for malaria diagnosis in Honduras is RDT. So it is better to compare with RDT too.

Answer. The reviewer is correct. In a more complete study, the ideal would have been to compare microscopy, Gazelle, PCR, and an RDT. However, the hospital that agreed to participate in the study in the city of Puerto Lempira does not perform RDTs and the diagnosis is based exclusively on the diagnosis of thick film, which, according to PAHO, remains the gold standard method for the diagnosis of malaria. Despite this limitation, the authors believe that our results are still valuable and contribute significantly to improving new diagnostic techniques.

  • PCR detected more cases which were missed by Gazalle or microscopy, what is the treatment given in those cases as all are fever cases.

Answer. The national malaria standard states that only those patients with a confirmed diagnosis of malaria should receive antimalarial treatment. Since the Gazelle results and even more so the PCR results were obtained a posteriori, the patients were probably treated symptomatically, which constitutes a serious problem for malaria control in the region.

  • How author can conclude that Gazelle is more sensitive in P. vivax infection as compared to the P. falciparum as it is not the study design. If author want to compare the species then sample size may calculated accordingly the prevalence of those species in the study area.

Answer. We would appreciate if the reviewer could tell us to which line he/she is referring. As far as the discussion, we are not making any conclusions we are just referring that there is a high percentage of missing infections by Gazelle and that it could be due to sensitivity differences.

  • Overall manuscript need extensive rewrite in every section.

Answer. We are sorry that the reviewer found the writing of the manuscript to be deficient. We are happy to improve it if you accurately point out the errors found.

Reviewer 2 Report

Title: Field evaluation of a hemozoin-based malaria diagnostic device in Puerto Lempira, Honduras

First of all, I would like to thank the editorial team for giving me the opportunity to review this paper. The authors evaluated the diagnostic performances and practicality of a hemozoin-based device for P. falciparum (Pf) and P. vivax (Pv) detection in field conditions in Honduras, a Latin America country where these two species are highly prevalent. The work by Fontecha and colleagues is well written, the results are interesting, and I think the paper reached quality standards for publication to “DIAGNOSTICS”. However, before it, the authors should revise the paper based on some major and minor questions that you can find below.

Major revisions

Introduction

  • Lines 77: What do you mean by “uninfected individuals”? The presence of hemozoin may mean past or present infection. Regarding past infected individuals, do you consider them as infected or uninfected?

Materials and Methods

  • Lines 91-95: Detailed description of the study area in terms of climate, malaria mosquito species, and epidemiology of Plasmodium species in deeply lacking in the text.
  • Lines 135-138: The authors have to clearly state the formula used for setting sample size. There are several good previous publications that addressed this aspect for diagnostic performance studies, depending on the number of techniques evaluated

Results

  • Table 5: Give clear details on components used for determining cost of each technique

Discussion

  • Lines 343-345: Given this statement the authors should put a comment on GazelleTM detection results for submicroscopic infections

Minor revisions

Abstract

  • Lines 21-23: The authors should present performances of GazelleTM (sensitivity, specificity, and kappa) in comparison to microscopy and nPCR with statistical significance.

Introduction

  • Lines 39-41: Add a bibliographic reference.
  • Lines 46-47: Light microscopy (LM) is also useful for evaluating malaria severity through parasitemia quantification. Please add this aspect.
  • Line 44: Change « light microscopy » to « light microscopy (LM) ».
  • Line 48: Change « microscopy » to « LM ». also use the abbreviation in the rest of paper
  • Line 60: You said « nor do they discriminate species». I am not agreeing with this statement as some RDTs are designed to discriminate between Pf and Pv. Reformulate the sentence

Materials and Methods

  • Lines 97: Change « subjects » to « population ».
  • Figure 1. Scale is missing in the maps.
  • Lines 151-165: A photograph/image of the GazelleTM is required.
  • Lines 164-165: You stated that the GazelleTM device is unable to differentiate between Plasmodium So, why you present the results of this device for Pf and Pv separately?
  • Line 241: The authors should give clear interpretation of the kappa index as done on lines 233-235 for AUC
  • Why the authors did not include RDT in the study? You stated in the introduction that RDTs and LM are commonly used in Honduras.

Results

  • Table 4. The authors have to compare LM and GazelleTM statistically for the parameters (McNemar chi square test) in order to support their statement on comparability of the both techniques’ results
  • Figure 2. It should be “1 – specificity” on the x-axis of the graph instead of “specificity”
  • Lines 280-281: Clearly give an interpretation of AUC for LM (Very good, AUC = 0.82) and GazelleTM (Good, AUC = 0.79).

Discussion

  • Lines 323-334. These sentences can be shortened.

Author Response

Major revisions

  • Lines 77: What do you mean by “uninfected individuals”? The presence of hemozoin may mean past or present infection. Regarding past infected individuals, do you consider them as infected or uninfected?

Answer. “Uninfected individuals” refers to individuals who are not infected by Plasmodium The reviewer indicates that the presence of hemozoin could indicate a past infection, however, the literature indicates that the spleen and liver eliminate hemozoin efficiently, so the probability of detecting it in peripheral blood is minimal (Delahunt et al 2014). Consequently, hemozoin is a good and highly specific biomarker of infection (de Melo et al 2021). However, Arndt et al showed that past malaria infections are associated with residual levels of hemozoin, especially in individuals infected within the last two weeks. Thus, we have added the phrase: "Hemozoin is a good biomarker because it is generally absent in uninfected individuals"... to avoid generalizing and considering the cases in which there is remnant hemozoin.

  • Materials and Methods. Lines 91-95: Detailed description of the study area in terms of climate, malaria mosquito species, and epidemiology of Plasmodiumspecies in deeply lacking in the text.

Answer. The following paragraph has been added: “La Moskitia is an isolated territory located in eastern Honduras (Figure 1) that contributed to 98% of national malaria cases last year. Two species of human Plasmodia coexist in the region, P. falciparum and P. vivax [30]. The most common vector species in the country and in La Moskitia is Anopheles albimanus [31]. The geographic isolation of the Honduran Moskitia is caused by multiple lagoons and a thick strip of forest that limits land access. Around this system of lagoons of 3,700 square kilometers live more than 24,000 people far from each other. The remoteness and the difficulties of access limit the development of the communities with respect to the rest of the country, which is reflected in the low health and education indicators, with a human poverty rate of 53%. The mangrove forest is the dominant coastal ecosystem with more than 23,000 hectares, which privileges fishing as the main means of subsistence, and favor the development of malaria vector mosquitoes [32].

  • Lines 135-138: The authors have to clearly state the formula used for setting sample size. There are several good previous publications that addressed this aspect for diagnostic performance studies, depending on the number of techniques evaluated

Answer. We are stating the reference for the formula and adding additional details regarding the calculation.

  • Table 5: Give clear details on components used for determining cost of each technique

Answer. The following paragraph was added: “The cost of taking the blood sample and preparing the thick smear for microscopy was included. The cost of sampling, cartridges and buffers for GazelleTM, filter paper, semi-automatic DNA extraction, amplification reactions and electrophoresis for PCR were considered”.

  • Lines 343-345: Given this statement the authors should put a comment on GazelleTMdetection results for submicroscopic infections

Answer. Done.

Minor revisions

  • Lines 21-23: The authors should present performances of GazelleTM(sensitivity, specificity, and kappa) in comparison to microscopy and nPCR with statistical significance.

Answer. Done.

  • Lines 39-41: Add a bibliographic reference.

Answer. Done.

  • Lines 46-47: Light microscopy (LM) is also useful for evaluating malaria severity through parasitemia quantification. Please add this aspect.

Answer. The sentence: “in addition to being useful to assess the severity of malaria through quantification of parasitaemia” has been added.

  • Line 44: Change « light microscopy » to « light microscopy (LM) ». Line 48: Change « microscopy » to « LM ». also use the abbreviation in the rest of paper

Answer. We have changed to LM in the document.

  • Line 60: You said « nor do they discriminate species». I am not agreeing with this statement as some RDTs are designed to discriminate between Pf and Pv. Reformulate the sentence

Answer. We have removed the phrase.

  • Materials and Methods. Lines 97: Change « subjects » to « population ».

Answer. Done.

  • Figure 1. Scale is missing in the maps.

Answer. Scale has been added in figure 1.

  • Lines 151-165: A photograph/image of the GazelleTM is required.

Answer. A new figure (2) has been included showing the Gazelle device.

  • Lines 164-165: You stated that the GazelleTM device is unable to differentiate between Plasmodium So, why you present the results of this device for Pf and Pv separately?

Answer. It is correct. Gazelle is able to detect Plasmodium infections but is not yet able to differentiate species. The reasons why we decided to separate the analysis by parasite species are: (1) to try to understand if there were differences in the ability of the device to diagnose between species compared to PCR, and (2) to compare microscopy with PCR, which in both cases differentiate between species.

  • Line 241: The authors should give clear interpretation of the kappa index as done on lines 233-235 for AUC

Answer. “The kappa index result be interpreted as follows: values ≤ 0 indicate no agreement and 0.01–0.20 as none to slight, 0.21–0.40 as fair, 0.41– 0.60 as moderate, 0.61–0.80 as substantial, and 0.81–1.00 as almost perfect agreement”.

  • Why the authors did not include RDT in the study? You stated in the introduction that RDTs and LM are commonly used in Honduras.

Answer. The reviewer is correct. In a more complete study, the ideal would have been to compare microscopy, Gazelle, PCR, and an RDT. However, the hospital that agreed to participate in the study in the city of Puerto Lempira does not perform RDTs and the diagnosis is based exclusively on the diagnosis of thick film, which, according to PAHO, remains the gold standard method for the diagnosis of malaria. Despite this limitation, the authors believe that our results are still valuable and contribute significantly to improving new diagnostic techniques.

  • Table 4. The authors have to compare LM and GazelleTM statistically for the parameters (McNemar chi square test) in order to support their statement on comparability of the both techniques’ results

Answer. McNemar chi square test was conducted, and the results validated our statement (p=0.6831). Results were added in the paper.

  • Figure 2. It should be “1 – specificity” on the x-axis of the graph instead of “specificity”

Answer. Thanks, Figure has been modified.

  • Lines 280-281: Clearly give an interpretation of AUC for LM (Very good, AUC = 0.82) and GazelleTM (Good, AUC = 0.79).

Answer. Done.

  • Lines 323-334. These sentences can be shortened.

Answer. We have revised the wording of the sentences and feel that shortening them may sacrifice the clarity of the text.

Reviewer 3 Report

Dear authors,

the manuscript is well written.

I have a few comments :

Did you perform a RDT study in parallel with gazelle and other techniques ? RDT is most used now in endemic areas as you said in introduction. I really think that if you use gazelle in this study to perfrom diagnosis in addition or instead of RDT, RDT must have been used in parallel to compare outcome.

There is not enough conclusion about the really poor outcome in P. falciparum diagnosis by gazelle, which is lacking a lot ! Missing more than 50% of the parasites is not possible for diagnosis.

I'm really surprised by the same numbers for microscopy, a lot of parasites were missed from microscopy diagnosis. Why ?

About the ethics, the explanations inside the manuscript must be inside the ethics statement in the end of the paper because it only says here :

"Informed Consent Statement: Informed consent was obtained from all subjects involved in the study." --> I really doubt that a 1 month old baby is giving his consent alone.

Author Response

  • Did you perform a RDT study in parallel with gazelle and other techniques ? RDT is most used now in endemic areas as you said in introduction. I really think that if you use gazelle in this study to perfrom diagnosis in addition or instead of RDT, RDT must have been used in parallel to compare outcome.

Answer. The reviewer is correct. In a more complete study, the ideal would have been to compare microscopy, Gazelle, PCR, and an RDT. However, the hospital that agreed to participate in the study in the city of Puerto Lempira does not perform RDTs and the diagnosis is based exclusively on the diagnosis of thick film, which, according to PAHO, remains the gold standard method for the diagnosis of malaria. Despite this limitation, the authors believe that our results are still valuable and contribute significantly to improving new diagnostic techniques.

  • There is not enough conclusion about the really poor outcome in  falciparum diagnosis by gazelle, which is lacking a lot ! Missing more than 50% of the parasites is not possible for diagnosis.

Answer. We agree with the reviewer. In La Moskitia an interesting phenomenon is taking place in this stage prior to elimination: very low parasitaemia! Our next step will be to conduct a deeper study evaluating this phenomenon using qPCR to try to better understand it and help decision-makers. So far, we cannot speculate further on this, and can only suggest what is mentioned in the next paragraph: “On the other hand, GazelleTM missed 40% of vivax infections but 52.6% of P. falciparum infections, which could be due to a difference in sensitivity for both species. Some authors have shown that P. falciparum infections tend to go unnoticed by magneto-optical detection of hemozoin [44]. Since the concentration of hemozoin in the peripheral bloodstream depends in part on the circulating stages of the parasite, and because ring immature trophozoites predominate in P. falciparum infections in the absence of other asexual stages, the amount of hemozoin is lower compared to P. vivax infection [48,49]”.

  • I'm really surprised by the same numbers for microscopy, a lot of parasites were missed from microscopy diagnosis. Why ?

Answer. Unfortunately, we cannot explain in this manuscript the reasons for the current phenomenon in Honduras and Nicaragua, however we will continue working to deepen it.

  • About the ethics, the explanations inside the manuscript must be inside the ethics statement in the end of the paper because it only says here: "Informed Consent Statement: Informed consent was obtained from all subjects involved in the study." --> I really doubt that a 1 month old baby is giving his consent alone.

Answer. The statement has been modified as follows: “Informed consent was obtained from all subjects involved in the study and/or their legal guardians in the case of being minors”.

Round 2

Reviewer 1 Report

Author responded almost all the query however i still not comfortable with the sample size as it is too small.